# A randomized, controlled Phase 1b trial of the *Sm*-TSP-2 Vaccine for intestinal schistosomiasis in healthy Brazilian adults living in an endemic area

**David J. Diemert**[1,2]*, **Rodrigo Correa-Oliveira**[3], **Carlo Geraldo Fraga**[3], **Frederico Talles**[3], **Marcella Rezende Silva**[3], **Shital M. Patel**[4], **Shirley Galbiati**[5], **Jessie K. Kennedy**[5], **Jordan S. Lundeen**[5], **Maria Flavia Gazzinelli**[3], **Guangzhao Li**[2], **Lara Hoeweler**[2], **Gregory A. Deye**[6], **Maria Elena Bottazzi**[7], **Peter J. Hotez**[7], **Hana M. El Sahly**[4], **Wendy A. Keitel**[4], **Jeffrey Bethony**[2], **Robert L. Atmar**[4]

1 Department of Medicine, School of Medicine and Health Sciences, The George Washington University, Washington DC, United States of America, 2 Department of Microbiology, Immunology and Tropical Medicine, School of Medicine and Health Sciences, The George Washington University, Washington DC, United States of America, 3 Instituto René Rachou, Fundação Oswaldo Cruz em Minas Gerais, Belo Horizonte, Minas Gerais, Brazil, 4 Departments of Molecular Virology & Microbiology and Medicine, Baylor College of Medicine, Houston, Texas, United States of America, 5 The Emmes Company, LLC, Frederick, Maryland, United States of America, 6 Division of Microbiology and Infectious Diseases (DMID), National Institutes of Allergy and Infectious, Diseases (NIAID), National Institutes of Health (NIH), United States of America, 7 Texas Children's Hospital Center for Vaccine Development, Department of Pediatrics, Baylor College of Medicine, Houston, Texas, United States of America

* ddiemert@gwu.edu

**Data Availability Statement:** All relevant clinical data are within the manuscript and its Supporting Information files. Raw antibody data have been

## Abstract

### Background

Recombinant *Schistosoma mansoni* Tetraspanin-2 formulated on Alhydrogel (*Sm*-TSP-2/Alhydrogel) is being developed to prevent intestinal and hepatic disease caused by *S. mansoni*. The tegumentary *Sm*-TSP-2 antigen was selected based on its unique recognition by cytophilic antibodies in putatively immune individuals living in areas of ongoing *S. mansoni* transmission in Brazil, and preclinical studies in which vaccination with *Sm*-TSP-2 protected mice following infection challenge.

### Methods

A randomized, observer-blind, controlled, Phase 1b clinical trial was conducted in 60 healthy adults living in a region of Brazil with ongoing *S. mansoni* transmission. In each cohort of 20 participants, 16 were randomized to receive one of two formulations of *Sm*-TSP-2 vaccine (adjuvanted with Alhydrogel only, or with Alhydrogel plus the Toll-like receptor-4 agonist, AP 10–701), and 4 to receive Euvax B hepatitis B vaccine. Successively higher doses of antigen (10 µg, 30 µg, and 100 µg) were administered in a dose-escalation fashion, with progression to the next dose cohort being dependent upon evaluation of 7-day safety data after all participants in the preceding cohort had received their first dose of vaccine. Each participant received 3 intramuscular injections of study product at intervals of 2 months and was

uploaded to the Dryad Digital Repository (unique digital object identifier: 10.5061/dryad.612jm647c).

**Funding:** This project has been funded with Federal funds from the National Institute of Allergy and Infectious Diseases (NIAID). Research was supported by a NIAID DMID Vaccine and Treatment Evaluation Unit (VTEU) award to Baylor College of Medicine (Contract No. HHSN27220130015I) and a NIAID DMID Statistical and Data Coordinating Center for Clinical Research in Infectious Diseases (SDCC) award to Emmes Corporation (Contract No. HHSN75N93021C00012). The funders had no role in study design, data collection and analysis, decision to publish, or preparation of the manuscript.

**Competing interests:** The authors declare the following financial interests/personal relationships which may be considered as potential competing interests: PH, MEB, DJD and JB are patent holders for a multivalent anthelminthic vaccine including schistosomiasis. GAD is an employee of the Sponsor, NIAID. RLA, RC-O, CGF, FT, MR, SG, JKK, JSL, MFG, GL, LH, WAK, SP, HMES: None.

followed for 12 months after the third vaccination. IgG and IgG subclass antibody responses to *Sm*-TSP-2 were measured by qualified indirect ELISAs at pre- and post-vaccination time points through the final study visit.

## Results

*Sm*-TSP-2/Alhydrogel administered with or without AP 10-701 was well-tolerated in this population. The most common solicited adverse events were mild injection site tenderness and pain, and mild headache. No vaccine-related serious adverse events or adverse events of special interest were observed. Groups administered *Sm*-TSP-2/Alhydrogel with AP 10–701 had higher post-vaccination levels of antigen-specific IgG antibody. A significant dose-response relationship was seen in those administered *Sm*-TSP-2/Alhydrogel with AP 10–701. Peak anti-*Sm*-TSP-2 IgG levels were observed approximately 2 weeks following the third dose, regardless of *Sm*-TSP-2 formulation. IgG levels fell to low levels by Day 478 in all groups except the 100 µg with AP 10–701 group, in which 57% of subjects (4 of 7) still had IgG levels that were $\geq$4-fold higher than baseline. IgG subclass levels mirrored those of total IgG, with IgG1 being the predominant subclass response.

## Conclusions

Vaccination of adults with *Sm*-TSP-2/Alhydrogel in an area of ongoing *S. mansoni* transmission was safe, minimally reactogenic, and elicited significant IgG and IgG subclass responses against the vaccine antigen. These promising results have led to initiation of a Phase 2 clinical trial of this vaccine in an endemic region of Uganda.

## Trial registration

NCT03110757.

### Author summary

Infection caused by *Schistosoma mansoni* is a major neglected tropical disease with significant associated morbidity. New tools, such as vaccines, are needed due to the inadequacy of current control strategies. Tetraspanin-2 of *S. mansoni* (*Sm*-TSP-2) is one of the lead vaccine candidates for hepatic/intestinal schistosomiasis. Antibodies induced by this vaccine are postulated to interfere with the development of the tegument of adult *S. mansoni* worms, thereby impairing their development and survival. We conducted a Phase 1 trial of recombinant *Sm*-TSP-2 adjuvanted with Alhydrogel in 60 healthy adults living in Brazil. Each participant received three vaccinations every 2 months by intramuscular injection of the vaccine administered with or without an aqueous solution of the Toll-like receptor-4 agonist, Glucopyranosyl Lipid A (AP 10–701). *Sm*-TSP-2/Alhydrogel was well tolerated in schistosomiasis-exposed adults; no vaccine-related severe or serious adverse events were observed. Antigen-specific IgG antibodies were induced in a dose-dependent fashion with increasing levels observed after each vaccination. The addition of AP 10–701 to the vaccine resulted in significantly higher antibody responses. Based on these results, the vaccine has been advanced into a Phase 2 clinical trial in an endemic region of Africa.

## Introduction

Approximately 240 million people in 54 countries are infected with *Schistosoma* blood flukes, with up to 700 million people at risk of infection [1]. Although acute infection can cause significant morbidity and even death due to Katayama fever, the greater global burden of disease due to this parasite results from chronic infection that can lead to life-threatening complications such as portal hypertension associated with *S. mansoni* or bladder obstruction, kidney failure, and bladder cancer associated with *S. haematobium* [2]. The intestinal and hepatic schistosomiasis caused by *S. mansoni* accounts for approximately one-third of schistosomiasis cases and almost half of deaths worldwide [3]. *S. mansoni* is the only schistosome species endemic in the Americas, with the largest number of infections in this region occurring in Brazil [4].

Currently, the control of schistosomiasis relies on mass drug administration of praziquantel primarily to school-age children 5 to 15 years of age, who tend to have the highest intensity of infection and disease burden. However, control programs based on mass chemotherapy are complicated by rapid and frequent re-infection following treatment. These programs are difficult and expensive to maintain, as has been highlighted by the interruption of many of these programs during the Severe Acute Respiratory Syndrome Coronavirus 2 (SARS-CoV-2) pandemic [5]. Additionally, significant morbidity can occur if regular periodic treatments are interrupted [6,7]. Furthermore, reduced responsiveness to repeated treatment with praziquantel suggests potential emerging drug resistance, a concern given the lack of alternative therapies [8–10]. Alternative tools such as an effective anti-schistosomal vaccine are therefore needed to adequately control, and ultimately eliminate, the burden of disease caused by this parasitic infection.

Several anti-schistosomal vaccines are currently in preclinical or clinical development [11,12]. Recombinant *S. mansoni* Tetraspanin-2 (*Sm*-TSP-2) formulated on Alhydrogel, an aluminum hydroxide salt adjuvant, is being developed to prevent heavy infections with *S. mansoni*, given that morbidity is related to infection intensity [13]. Schistosomes express a family of integral transmembrane proteins that locate to the epithelial syncytium forming the body wall, or tegument, of several stages of the schistosome life cycle, and are considered essential to the formation, maintenance, and turnover of the transient outer surface of this helminth [14,15]. Evidence for this comes from *in vitro* RNA interference experiments in which schistosomula were exposed to *Sm*-tsp-2 double-stranded RNA, resulting in significant reductions in *Sm*-tsp-1 transcription levels and significantly thinner and more vacuolated tegument with a morphology consistent with a failure of tegumentary invaginations to close. The most compelling data come from field studies conducted in the same region as the Phase 1b trial reported herein, which demonstrated that *Sm*-TSP-2 was uniquely recognized by cytophilic IgG1 and IgG3 antibodies from putatively resistant individuals at levels significantly higher than age-, sex-, and exposure-matched chronically infected controls [16]. Finally, preclinical testing showed that vaccination with this antigen protected mice in a challenge model of *S. mansoni* infection [16].

*Sm*-TSP-2/Alhydrogel administered with or without an aqueous formulation of the Toll-like receptor (TLR)-4 agonist, glucopyranosyl lipid A (AP 10–701), was tested in 72 healthy adults without a history of schistosomiasis in a Phase 1a trial in the United States. It was shown to be safe and well-tolerated, and it induced significant antigen-specific serum IgG responses [17]. However, since safety or immunogenicity results may differ between naïve and previously exposed and/or infected individuals, a Phase 1b dose-escalation trial was conducted in a region of Brazil where *S. mansoni* is endemic, as described herein. Potential study participants were screened for the presence of IgE antibodies to recombinant *Sm*-TSP-2 prior to enrollment,

given the experience with the *Necator americanus* Ancylostoma Secreted Protein-2 (*Na*-ASP-2) hookworm vaccine, in which previously infected individuals who had IgE antibodies to the vaccine antigen developed generalized urticaria upon vaccination [18].

The primary and secondary objectives of this study were to evaluate the safety and immunogenicity of *Sm*-TSP-2/Alhydrogel administered with or without AP 10–701 to healthy Brazilian adults residing in an area of ongoing *S. mansoni* transmission compared to the comparator hepatitis B vaccine. This report is the first to present results of the *Sm*-TSP-2/Alhydrogel vaccine for intestinal and hepatic schistosomiasis when evaluated in individuals living in a *S. mansoni*-endemic area.

## Methods

### Ethics statement

The study was approved by the ethics committees of the Federal University of Minas Gerais, the George Washington University, Baylor College of Medicine, and the Brazilian Ministry of Health. Written informed consent was obtained from volunteers who successfully completed a questionnaire assessing comprehension of study procedures and risks before the initiation of study procedures. The trial has been registered at clinicaltrials.gov (https://clinicaltrials.gov/ct2/show/NCT03110757).

### Study vaccines

The *Sm*-TSP-2 schistosomiasis vaccine evaluated in this clinical trial was made in compliance with current Good Manufacturing Practice at Aeras (Maryland, USA) as a recombinant protein adsorbed to Alhydrogel [19,20]. *Sm*-TSP-2/Alhydrogel consists of 0.1 mg recombinant *Sm*-TSP-2 adsorbed to 0.8 mg/mL Alhydrogel in 10 mM imidazole, 15% sucrose, 2 mM phosphate (pH 7.4 ± 0.1). Each dose concentration of recombinant protein that was tested in this trial (10 μg, 30 μg, and 100 μg) was administered by injecting an appropriate volume of *Sm*-TSP-2/Alhydrogel suspension. Doses of *Sm*-TSP-2/Alhydrogel were administered with or without the addition of 5 μg AP 10–701 (Infectious Diseases Research Institute, Seattle, WA). AP 10–701 was combined with *Sm*-TSP-2/Alhydrogel vaccine not more than 4 hours prior to administration.

The comparator was the Euvax B (LG Life Sciences, South Korea) recombinant hepatitis B vaccine containing 20 μg hepatitis B surface antigen and 0.5 mg aluminum per dose. Euvax B was supplied in multi-dose vials each containing enough vaccine to deliver ten-1.0 ml doses.

### Study site and population

This Phase 1b, randomized, observer-blind (within cohort), controlled, dose-escalation clinical trial was conducted in Americaninhas, Minas Gerais, Brazil, by a research team based at the Instituto René Rachou, part of the Fundação Oswaldo Cruz of the federal Brazilian Ministry of Health. Previous studies conducted in Americaninhas and the surrounding area have demonstrated ongoing *S. mansoni* transmission [21–23].

Sixty healthy male and non-pregnant, non-breastfeeding female adults aged 18 to 50 years, inclusive, were recruited. Exclusion criteria included positivity for IgE antibodies to *Sm*-TSP-2 by ELISA; clinically significant systemic disease; chronic infection with hepatitis B, hepatitis C, or HIV viruses; and history of allergy to any vaccine component. Other exclusions included current use of oral, parenteral, or high-dose inhaled corticosteroids or other immunosuppressive drugs; and receipt of an inactivated vaccine within the 2 weeks prior to the first study vaccination, or a live vaccine within the prior 30 days. Women of childbearing potential were

required to have a negative urine pregnancy test within 24 hours before each vaccination and to agree to use highly effective contraception from 30 days prior to the first injection until 1 month following the final administration of study product.

Infection with the intestinal helminths *Ascaris lumbricoides*, hookworm, *Strongyloides stercoralis*, *Trichuris trichiura*, *S. mansoni*, and *Taenia spp* was determined using the Kato Katz fecal thick smear microscopy technique on 2 samples collected on separate days from each potential participant during the screening period and at least 3 weeks prior to planned enrollment. If screening fecal exams demonstrated infection with an intestinal parasite, the individual was offered treatment with an appropriate medication prior to being enrolled in the study and before any vaccinations. Those positive for *S. mansoni* were treated with a single 60 mg/kg dose of praziquantel within 2 weeks but at least 10 days prior to the first vaccination.

## Clinical procedures

Participants were progressively enrolled into 3 cohorts. In each cohort of 20, participants were randomly assigned to receive *Sm*-TSP-2/Alhydrogel (n = 8), *Sm*-TSP-2/Alhydrogel with 5 μg AP 10–701 (n = 8), or the comparator vaccine (n = 4). Those randomized to receive *Sm*-TSP-2/Alhydrogel (with or without AP 10–701) were administered 10 μg, 30 μg, and 100 μg of antigen in cohorts 1, 2, and 3, respectively. Intramuscular injections of vaccine were administered on study days 1, 57, and 113 in the deltoid muscle, with successive vaccinations given in alternating arms. Participants were followed until approximately 12 months after the third vaccination (study day 478).

Within each cohort, an initial 5 participants (2 *Sm*-TSP-2/Alhydrogel, 2 *Sm*-TSP-2/Alhydrogel/AP 10–701, and 1 comparator vaccine) were enrolled, randomized, vaccinated, and had completed the study day 2 visit before enrolling the remainder of the cohort if no pre-defined pause criteria had been met. Furthermore, enrollment into the cohorts was staggered such that safety and reactogenicity data for the 7 days following the first vaccinations of cohorts 1 and 2 were reviewed by the investigators and the Sponsor's medical monitor prior to initiating cohort 2 and 3 vaccinations, respectively. Dose escalation decisions were evaluated according to protocol-defined criteria, with evidence of significant reactogenicity requiring an enrollment pause pending further review by the Safety Monitoring Committee prior to proceeding.

## Blinding

Participants, investigators, personnel performing study-related assessments following vaccine administration, and laboratory personnel performing antibody assays were blinded to study product assignment. Randomization was performed using an internet-based randomization system (Advantage eClinical, The Emmes Company, Rockville, MD). The randomization scheme was provided to unblinded study personnel (*i.e.*, individuals performing study vaccine preparation and administration). The unblinded study vaccine administrator was a study member trained to administer vaccines who also participated in dose preparation but was not involved in study-related safety assessments and had no participant contact for data collection following study vaccine administration.

## Clinical assessments

Reactogenicity was measured from the time of each study vaccination (study days 1, 57, 113) through 7 days after each study vaccination by the occurrence of solicited injection site and systemic reactogenicity events. Solicited symptoms included erythema, induration/swelling, pain, and tenderness as injection site reactions; and fever, chills, myalgia, arthralgia, nausea, vomiting, headache, dizziness, malaise, and fatigue as systemic reactions.

Unsolicited adverse events (AEs) were recorded from the time of each vaccination through 28 days after each study product administration. New-onset chronic medical conditions (NOCMCs), including AEs of special interest (AESI), and serious AEs (SAEs) were recorded from the time of the first study vaccination (day 1) through the final study visit. Active surveillance for AESIs of autoimmune (*e.g.*, systemic lupus erythematosus, autoimmune thyroiditis) and inflammatory (*e.g.*, inflammatory bowel disease) etiology was conducted throughout this clinical trial due to the use of the novel AP 10–701 adjuvant [24].

Participants were observed in the study clinic for at least 60 minutes after vaccination to assess immediate reactogenicity. They were subsequently assessed at 1, 3, 7, 14, and 28 days after each vaccination, and then at regular intervals until 12 months after the third injection. AEs were assigned severity of mild (easily tolerated), moderate (interfered with activities of daily living), or severe (prevented activities of daily living); causality in relation to study product was determined based on investigator judgement. Injection site swelling and erythema were assessed as small (25 to 50 mm in diameter), medium (51 to 100 mm), or large (>100 mm); whereas oral temperature was graded as mild (38.0°C to 38.4°C), moderate (38.5°C to 38.9°C), or severe (≥39.0°C).

Clinical laboratory evaluations were performed on the day of vaccination and 7 days later to measure alanine aminotransferase (ALT), creatinine, and complete blood count (CBC) including hemoglobin (Hgb), platelets, and white blood cells (WBC). Abnormal safety laboratory test results were assessed as mild, moderate, or severe according to standardized toxicity tables [25]. In addition, at each clinic visit concomitant medications were reviewed, vital signs were assessed, and a targeted physical exam was performed, if indicated. Two fecal samples were collected on different days for microscopic examination for helminth eggs at approximately study day 293.

## Laboratory methods

Immunoglobulin E (IgE) antibodies specific to *Sm*-TSP-2 were measured in serum samples collected during screening using a qualified indirect ELISA with a heterologous standard calibration curve produced by serially diluted purified human IgE (Abbiotec; Clonality: HE1

Cat. No.: 250203. Lot Number:2012283). In brief, 96-well plates were prepared with either 1 μg/mL recombinant *Sm*-TSP-2 (drug substance) diluted in 1X phosphate-buffered saline (PBS), 0.5 μg/mL purified human IgE serially diluted in 1X PBS to generate a standard calibration curve (SCC), or 0.125 μg/mL IgE as a positive control. Plates were incubated overnight at 2–8°C and then washed with PBST20 to remove any unbound IgE or *Sm*-TSP-2. All wells were then blocked with PBS and 5% Tween 20 with 5% bovine serum albumin (Blocking buffer, or BB). Serum samples were also diluted in 5% BB before being added to test plates in duplicate at a 1:25 dilution. Plates were sealed and then incubated overnight at 2–8°C. A horseradish per-oxidase (HRP)-conjugated secondary IgE antibody was then added to the plates at a 1:1000 dilution in 5% BB and plates were incubated for 2 hours at room temperature (18–25°C). Plates were then washed and streptavidin-HRP added to all wells at a 1:1000 dilution in 5% BB and incubated at room temperature for 30 minutes protected from direct light. Plates were washed a final time before being developed with *O*-phenylenediamine dihydrochloride (OPD) (Sigma Aldrich). After 30 minutes, the reaction was stopped by adding $H_2SO_4$ (Ricca). Plates were then immediately read at an Optical Density of 492nm ($OD_{492nm}$) using a SpectraMax Plus 384 Microplate Reader (Molecular Devices). $OD_{492nm}$ values were collected using SOFT-max GXP PRO version 4 with the arithmetic mean of duplicate ODs of the purified human IgE heterologously interpolated onto 4-parameter logistic (4-PL) models of the purified IgE to derive the Arbitrary Units (AU) of anti-*Sm*-TSP-2 IgE.

Immunoglobulin G (IgG) antibodies specific to *Sm*-TSP-2 were measured in serum using a qualified indirect ELISA [17,26]. A standard reference serum (SRS) for the assay was made by pooling sera collected from 5 high IgG responders to *Sm*-TSP-2 who participated in the phase 1a study of *Sm*-TSP-2/Alhydrogel in healthy adults [17]. In brief, 96-well microtiter plates (Nunc Polysorb) were coated with 5 μg/mL of recombinant *Sm*-TSP-2 and kept overnight at 2–8˚C. Plates were then washed and blocked with a buffer (BB) consisting of PBS and Tween 20 with 5% Bovine Serum Albumin (Fitzgerald). After decanting the BB off the plates, test sera were added in duplicate at a 1:4000 dilution. Plates were sealed and kept at 2–8˚C overnight, and on the following day they were decanted and washed again. HRP-conjugated secondary mouse anti-human IgG (Clone JDC-10 [Southern Biotech]) was added at a 1:2500 dilution (ThermoFisher). Plates were covered and maintained at room temperature for 2 hours, followed by a wash and development using OPD (Sigma Aldrich) and read after 30 minutes at $OD_{492nm}$ following the addition of 2 N $H_2SO_4$ using a SpectraMax Plus 384 Microplate Reader (Molecular Devices). $OD_{492}$ readings were collected using SOFTmax GXP PRO version 4, with the arithmetic mean of duplicates of test sera homologously interpolated onto the 4-PL model of the human SRS to derive the AUs of anti-*Sm*-TSP-2 IgG.

IgG subclass antibodies specific to *Sm*-TSP-2 were similarly measured in serum using qualified indirect ELISAs using heterologous interpolation to derive AUs of IgG1, IgG3, and IgG4 to *Sm*-TSP-2 [17,26]. Myeloma-derived purified human IgG1, IgG3, or IgG4 (catalog numbers: 16-090707-1M, 16-16-0909707-3, 16-16-090707-4M, respectively [Southern Biotech]) were adsorbed directly to 96-well Nunc Maxisorb microtiter plates (Cat No: 439454) in doubling dilutions to derive a SCC for each IgG subclass. Test sera were added at the following dilutions to each plate as described above IgG1 (1:50), IgG3 (1:50), and IgG4 (1:25). Plates were covered and incubated overnight at 2–8˚C. On the following day the plates were washed, and a HRP-conjugated secondary mouse anti-human IgG1 (Clone HP6070), IgG3 (Clone HP6050), or IgG4 (HP6025) were added to the plates at 1:1000. Plates were sealed and further incubated at room temperature for 2 hours and then washed and developed using OPD (Sigma Aldrich) and read at $OD_{492nm}$ following the addition of 2 N $H_2SO_4$ using a SpectraMax Plus 384 Microplate Reader (Molecular Devices). Values were collected using SOFTmax GXP PRO version 4 with the mean of duplicates of test sera heterologously interpolated onto the 4-PL model of the myeloma-derived purified human IgG subclasses to derive the AUs of IgG1, IgG3 and IgG4 antibodies against *Sm*-TSP-2 for TSP-2.

All antibody data from the ELISAs have been uploaded to the Dryad Digital Repository (unique digital object identifier: 10.5061/dryad.612jm647c) [27].

## Statistical methods

Safety data were summarized for the safety population consisting of all enrolled participants who received at least one dose of study product (*i.e.*, all 60 enrolled participants). Participants were summarized by vaccine group, with those who received comparator vaccine pooled across cohorts.

Any medical condition that was present at the time that the participant was screened was considered baseline and not reported as an AE, unless it worsened in severity or increased in frequency during the study. When calculating the incidence of AEs (*i.e.*, on a per participant basis), each participant was counted once, and any repetition of unsolicited AEs within a participant was ignored for events coded in the same category by MedDRA. Events thus summarized were coded to the highest severity observed.

The proportion of participants reporting at least 1 solicited AE was summarized by event, vaccine group, and maximum severity, along with the 95% Wilson score confidence interval

(CI) for the proportion. Event rates were compared between each vaccine group and the pooled comparator vaccine group using Fisher's Exact Test. The proportion of participants reporting at least 1 unsolicited AE was summarized by vaccine group for each vaccination and across all vaccinations. Clinical laboratory events were summarized by severity, time point, and vaccine group.

### Immunogenicity analysis

The immunology results described are restricted to the per-protocol immunogenicity population, which did not include all values from the 4 participants who did not receive all 3 doses of study product, as described below. For these participants, specimens collected after the missed vaccine dose were excluded from the immunogenicity analyses. All comparisons between *Sm*-TSP-2/Alhydrogel groups and the comparator vaccine group included pooled Euvax B recipients from all 3 cohorts.

IgG and 3 IgG subclass (IgG1, IgG3, and IgG4) antibodies to the *Sm*-TSP-2 protein were measured on each day of vaccination, 14 days post-vaccination, and approximately 3, 6, and 12 months after the final dose of study product. Statistical summaries of IgG and IgG subclass levels, including mean and standard deviation; geometric mean and 95% CIs; and median, minimum, and maximum, were calculated for each vaccine group and time point.

The $R^2$ for the 4-PL SCCs of SRS generated in SOFTmax GXP PRO should be above 0.6 to meet the specification of an acceptable relationship (dose response) between the dilutions and the ODs for the 4-PL log regression model. The linearity and parallelism of the SCCs were then assessed as part of the statistical quality control for ELISA. A fully specified logit-log model was utilized to linearize the sigmoidal shape of the SCCs and an Analysis of Variance (ANOVA) test was used to estimate the statistical similarity among the SCCs of each plate in GraphPad Prism.

For each individual and post-vaccination time point, the ratio of the post-vaccination antibody level to the baseline level was calculated (*i.e.*, the fold-rise). Fold-rise distributions were depicted graphically via box plots by vaccine group and time point. A seroresponse was defined as a ≥4-fold rise over baseline. Fold-rise values were summarized by geometric mean by vaccine group and time point, as were proportions of seroresponders. Proportions of seroresponders on study day 127 were compared between controls (comparator vaccine) and each vaccine dose level group using Fisher's exact test. An analysis of covariance (ANCOVA) test was performed on log-transformed day 127 IgG antibody levels to compare responses between individuals who were negative for schistosomiasis at screening and those who were positive, including baseline IgG level, *Sm*-TSP-2 dose level, and AP 10–701 coadministration status as covariates.

A post-hoc Cochran Armitage test of trend was performed to investigate a possible dose-response relationship in the proportion of total IgG seroresponders among participants receiving *Sm*-TSP-2/Alhydrogel without AP 10–701 on study day 127. The comparator group was included in the test with a dose level of zero. The test was repeated for those receiving *Sm*-TSP-2/Alhydrogel with AP 10–701, again including the comparator group with a dose level of zero.

No imputation methods were planned for missing values.

## Results

### Participant flow and baseline data

A total of 127 adults were screened and 60 were enrolled in the study (**Fig 1**). Of the 67 individuals who were not enrolled, 20 were not in good health, 14 had an acute or chronic medical condition, 10 were unable to understand and comply with study procedures, 22 were eligible but not enrolled, and 29 failed other eligibility criteria. Only 1 individual of the 126 tested had

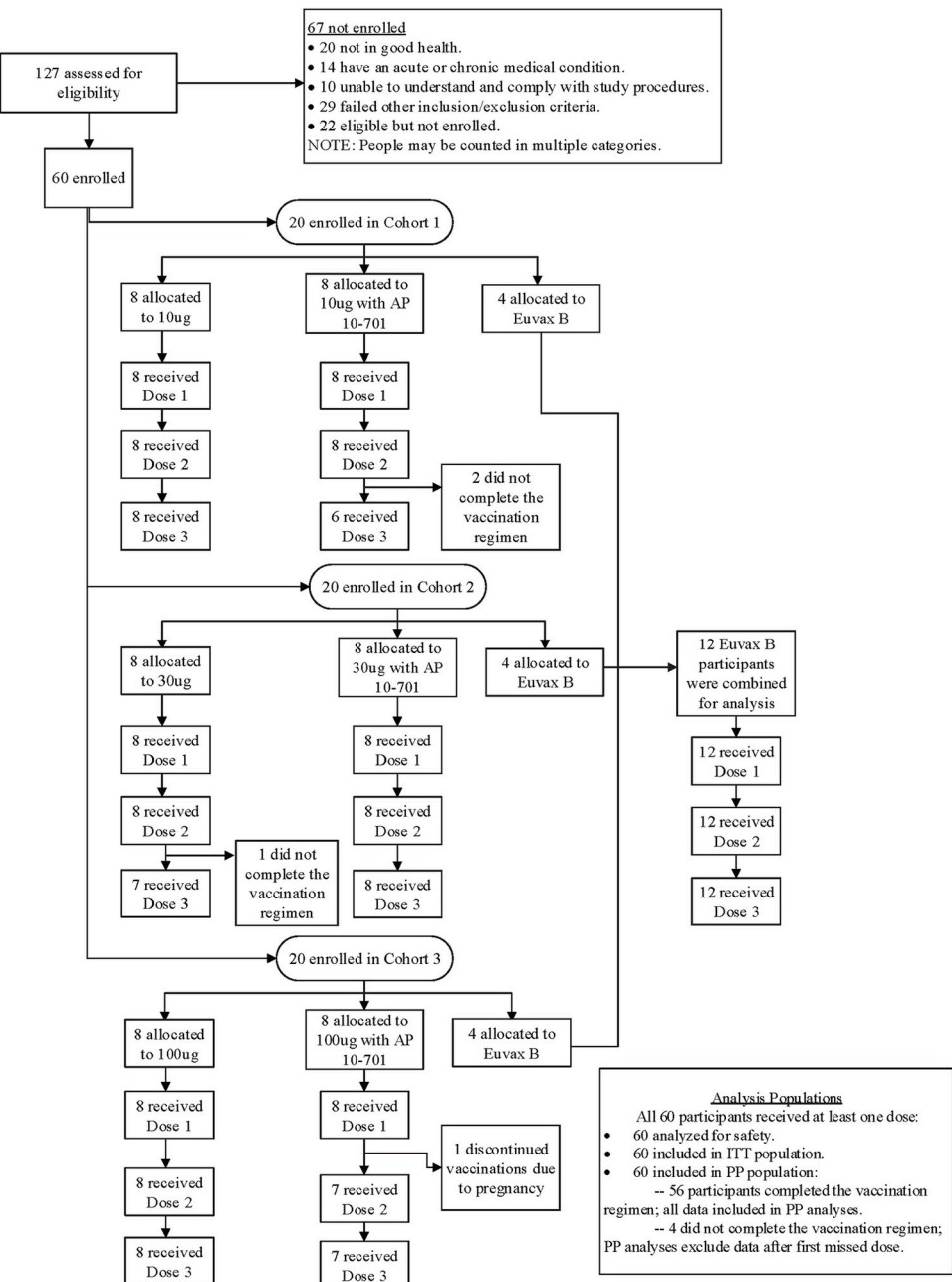

**Fig 1. CONSORT diagram: participant flow.**

detectable serum IgE antibodies specific for *Sm*-TSP-2. One participant who was randomized to the 30 μg *Sm*-TSP-2/Alhydrogel (without AP 10–701) group became ineligible after enrollment, when it was discovered that the subject had a previously undisclosed history of chronic idiopathic urticaria; the participant was discontinued at study day 112 prior to receiving the third dose of study vaccine.

All 60 enrolled participants received at least 1 dose of study vaccine, while 56 (93%) received all 3 planned doses (**Fig 1**): 2 participants in the 10 μg *Sm*-TSP-2/Alhydrogel with AP 10–701 group did not receive the third dose due to being lost to follow-up or withdrawn due to lack of availability; 1 participant in the 30 μg *Sm*-TSP-2/Alhydrogel group did not receive the third

**Table 1. Demographic and baseline characteristics of study participants by vaccine group.**

| | | 10µg *Sm*-TSP-2/ Alhydrogel (N = 8) | 10µg *Sm*-TSP-2/ Alhydrogel with AP 10–701 (N = 8) | 30µg *Sm*-TSP-2/ Alhydrogel (N = 8) | 30µg *Sm*-TSP-2/ Alhydrogel with AP 10–701 (N = 8) | 100µg *Sm*-TSP-2/ Alhydrogel (N = 8) | 100µg *Sm*-TSP-2/ Alhydrogel with AP 10–701 (N = 8) | Euvax B Vaccine (N = 12) | All (N = 60) |
|---|---|---|---|---|---|---|---|---|---|
| Sex | Male (n [%]) | 3 (38) | 3 (38) | 3 (38) | 4 (50) | 3 (38) | 1 (13) | 7 (58) | 24 (40) |
| | Female (n [%]) | 5 (63) | 5 (63) | 5 (63) | 4 (50) | 5 (63) | 7 (88) | 5 (42) | 36 (60) |
| Ethnicity | Hispanic/ Latino (n [%]) | 8 (100) | 8 (100) | 8 (100) | 8 (100) | 8 (100) | 8 (100) | 12 (100) | 60 (100) |
| Race | Multi-Racial (n [%]) | 8 (100) | 8 (100) | 8 (100) | 8 (100) | 8 (100) | 8 (100) | 12 (100) | 60 (100) |
| Age (years) | Mean (SD) | 36.9 (8.0) | 37.8 (5.6) | 34.9 (9.3) | 30.1 (8.7) | 30.4 (8.1) | 33.9 (8.8) | 34.7 (7.7) | 34.1 (8.1) |

Note: N = Number of participants in the Safety Population; SD = standard deviation.

dose due to an exclusionary condition (see above); and 1 participant in the 100 µg *Sm*-TSP-2/ Alhydrogel with AP 10–701 group did not receive the second or third doses due to pregnancy. Seven additional participants were lost to follow-up after receipt of all 3 doses of vaccine (2 in the 10 µg *Sm*-TSP-2/Alhydrogel group, 2 in the 30 µg *Sm*-TSP-2/Alhydrogel with AP 10–701 group, 1 in the 100 µg *Sm*-TSP-2/Alhydrogel group, and 2 in the comparator vaccine group), mainly due to moving away from the study site.

Demographic and baseline characteristics are summarized in **Table 1**. Overall, 60% of participants were female. The mean age for participants was 34.1 years (range: 18 to 47 years) with a median age of 35.0. Five of 60 (8.3%) study participants (1 in the 10 µg *Sm*-TSP-2/Alhydrogel group, 1 in the 30 µg *Sm*-TSP-2/Alhydrogel with AP 10–701 group, 2 in the 100 µg *Sm*-TSP-2/ Alhydrogel group, and 1 in the comparator group) were positive for *S. mansoni* by Kato Katz fecal thick smear at screening and were treated with praziquantel prior to vaccination. Of note, 2 (3.6%) study participants (1 each in the 10 µg *Sm*-TSP-2/Alhydrogel with AP 10–701 and 100 µg *Sm*-TSP-2/Alhydrogel with AP 10–701 groups) tested positive for *S. mansoni* at study day 293, neither of whom had been among the 5 who were treated at baseline. Fecal egg counts of participants who tested positive by Kato Katz thick smear at either screening or day 293 are shown in **S1 Table**.

## Safety results

None of the study participants in the safety population discontinued the study due to an AE, and no AESIs, NOCMCs, or deaths occurred during the clinical trial. One participant who received 30 µg *Sm*-TSP-2/Alhydrogel with AP 10–701 experienced a severe (grade 3), unsolicited SAE that was assessed as being unrelated to study product. This participant developed new-onset lower abdominal pain and vomiting starting 1 day following administration of the third dose of study vaccine. Subsequent investigations, including an abdominal ultrasound, revealed right ureterolithiasis with mild hydronephrosis and hydroureter. After failure of medical management, admission to hospital for endoscopic rigid ureterorenal lithotripsy with placement of a ureteral catheter led to complete symptom resolution.

## Solicited AEs

Out of 60 participants in the study, 45 (75%) experienced at least 1 solicited symptom, with 41 (68%) and 25 (42%) having at least 1 solicited injection site or systemic event, respectively

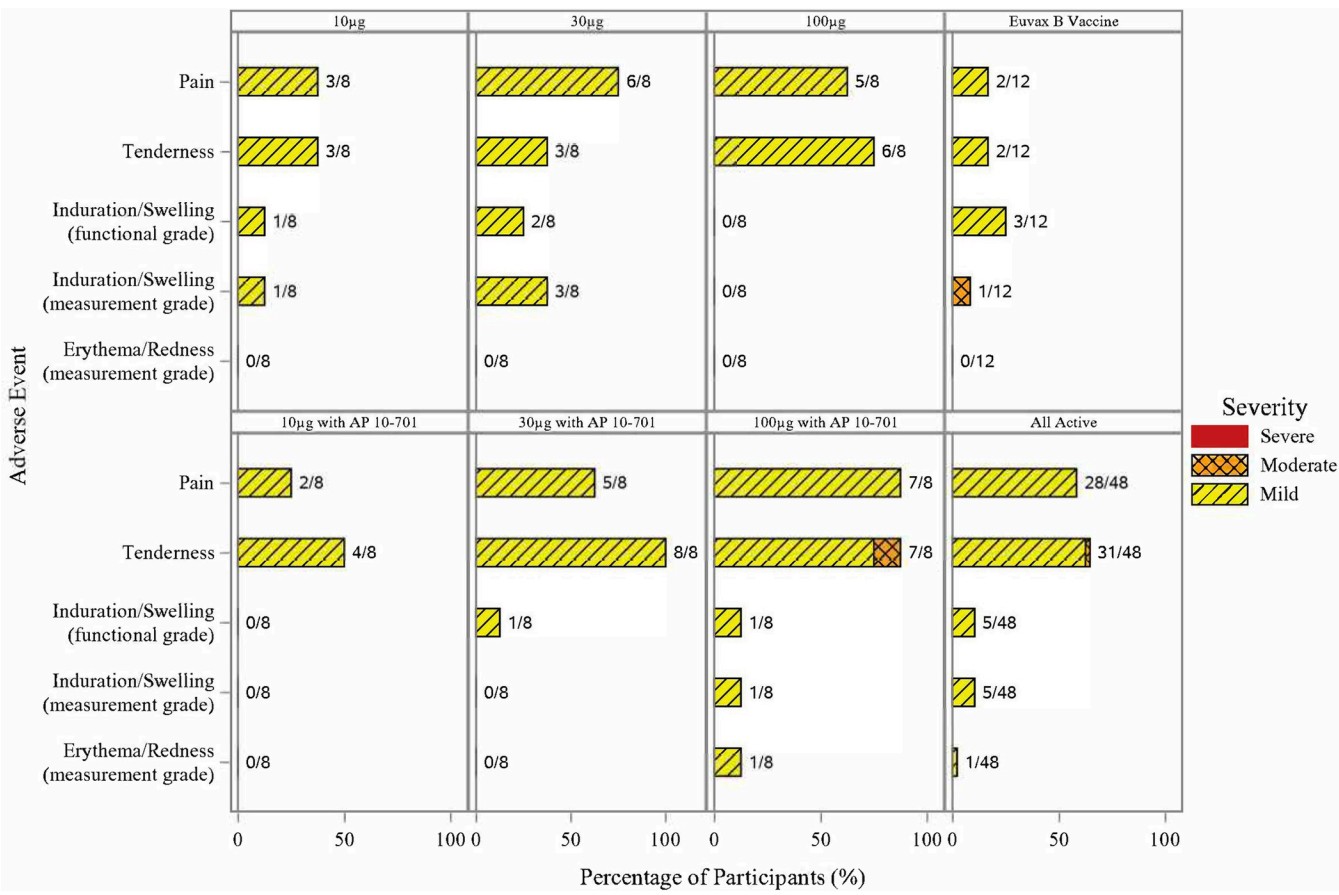

**Fig 2. Number and percentage of participants experiencing solicited injection site events after any dose of vaccine by event, maximum severity, and vaccine group.**

(**Figs 2** and **3**). Among all participants who received *Sm*-TSP-2/Alhydrogel with or without AP 10–701, the most common solicited systemic symptom was headache (17/48, 35%), whereas the most common solicited injection site symptom was tenderness (31/48, 65%).

No severe solicited AEs were observed in any study participant. In the combined group of participants who received any dose of *Sm*-TSP-2/Alhydrogel with or without AP 10–701, moderate systemic symptoms included headache (3/48, 6%), arthralgia (2/48, 4%), nausea (2/48, 4%), vomiting (1/48, 2%), fatigue (1/48, 2%), and myalgia (1/48, 2%). One (2%) moderate severity injection site symptom (tenderness) was observed after the first injection in a participant who received 100 μg *Sm*-TSP-2/Alhydrogel with AP 10–701. The most common mild systemic symptom reported in those vaccinated with *Sm*-TSP-2/Alhydrogel with or without AP 10–701 was headache (14/48, 29%). The most frequent mild injection site reactions were tenderness (30/48, 63%) and pain (28/48, 58%). In those who received *Sm*-TSP-2/Alhydrogel (with or without AP 10–701), 38 out of 48 participants reported injection site reactions that started on the day of vaccination or the following day and resolved by a median of 3.5 days post-vaccination (range, 1 to 17 days), whereas 21 out of 48 participants reported systemic reactions that lasted a median of 2.0 days (range, 1 to 10 days) (**S2 Table**). Reactogenicity occurred with similar frequency after the first, second, and third doses of vaccine.

A significantly greater percentage of participants experienced solicited events in the pooled group who received any dose of *Sm*-TSP-2/Alhydrogel with or without AP 10–701 (N = 48)

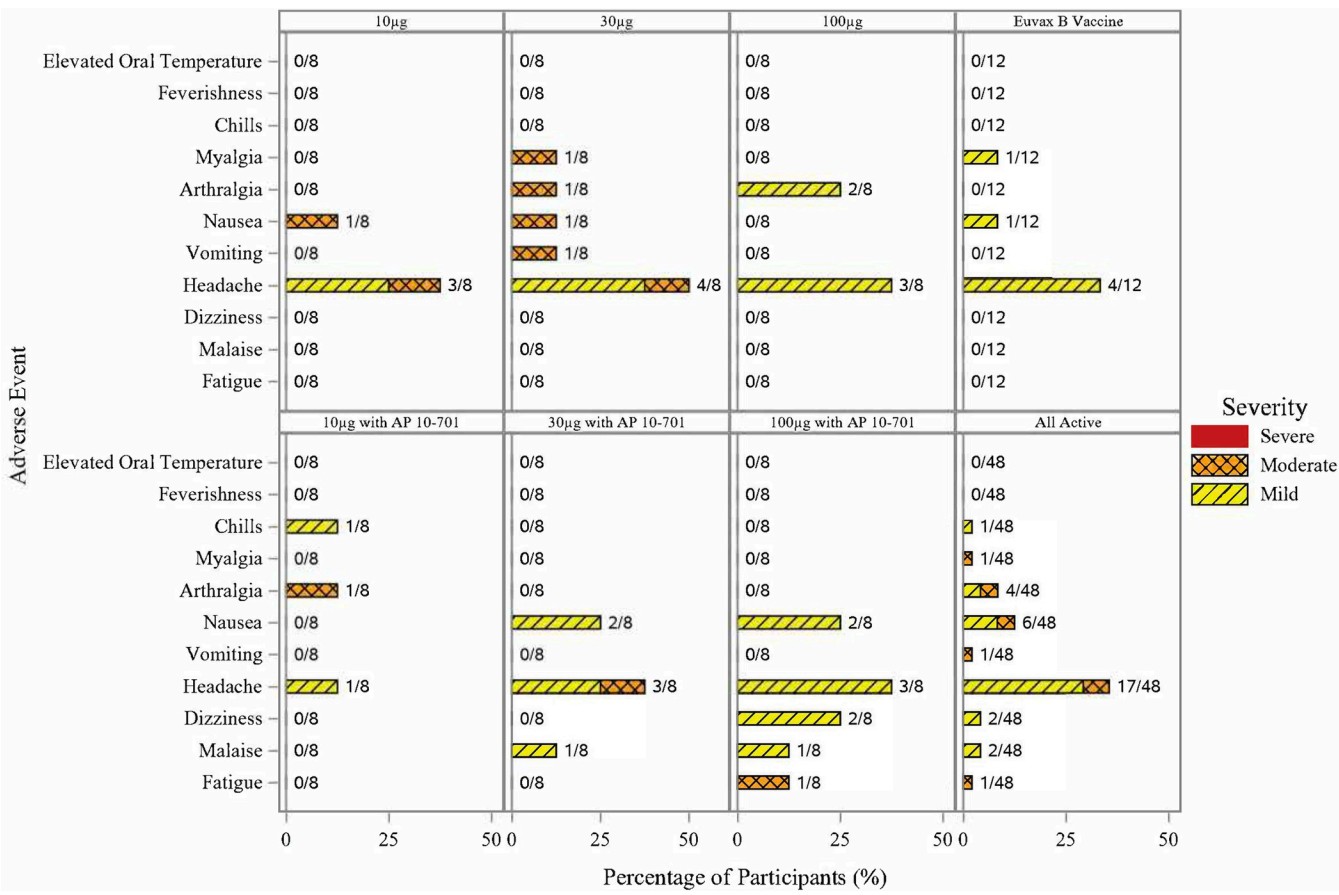

**Fig 3. Number and percentage of participants experiencing solicited systemic events after any dose of vaccine by event, maximum severity, and vaccine group.**

compared to the pooled comparator vaccine group (N = 12) (85% *vs.* 33%, *p*<0.001), primarily due to a significantly greater proportion of *Sm*-TSP-2/Alhydrogel recipients who had injection site symptoms compared to comparator vaccine (79% *vs.* 25%, *p*<0.001). There were no significant differences in the occurrence of systemic symptoms between the pooled *Sm*-TSP-2/Alhydrogel and comparator groups, or between the comparator vaccine group and any individual *Sm*-TSP-2/Alhydrogel group. Significantly greater proportions of participants experienced any solicited AE in the 30 μg *Sm*-TSP-2/Alhydrogel (88% *vs.* 33%, *p* = 0.028), 30 μg *Sm*-TSP-2/ Alhydrogel with AP 10–701 (100% *vs.* 33%, *p* = 0.005), and 100 μg *Sm*-TSP-2/Alhydrogel with AP 10–701 (100% *vs.* 33%, *p* = 0.005) groups compared to the comparator group. A higher proportion of injection site symptoms was observed in those vaccinated with 30 μg *Sm*-TSP-2/ Alhydrogel with AP 10–701 (100% *vs.* 25%, *p* = 0.001) and 100 μg *Sm*-TSP-2/Alhydrogel with AP 10–701 (100% *vs.* 25%, *p* = 0.001) compared to those vaccinated with comparator vaccine. The proportion of participants who experienced an injection site reaction was not significantly different between those who received *Sm*-TSP-2/Alhydrogel without AP 10–701 and those who received the vaccine with AP 10–701 (*p* = 0.3).

## Unsolicited AEs

Forty-eight participants experienced 96 non-serious unsolicited AEs, most of which were mild in severity. Three out of 60 participants (5%) experienced at least 1 unsolicited AE of mild or

moderate severity that was assessed as being related to study vaccine by the study investigators. These 3 participants experienced 7 (5 mild, 2 moderate) vaccine-related unsolicited AEs. Of these, 1 (moderate pruritus at the injection site) occurred after the third injection in a participant who received 100μg *Sm*-TSP-2/Alhydrogel without AP 10–701, whereas the remaining 6 (2 episodes of mild injection site pruritus, 1 episode of moderate injection site pruritus, 1 episode of mild injection site hematoma, 1 episode of mild injection site papular rash, and 1 episode of mild abdominal pain) occurred in 2 participants in the comparator vaccine group. There were no vaccine-related severe unsolicited AEs.

In the pooled group of participants who received 1 of the 3 dose concentrations of *Sm*-TSP-2/Alhydrogel with or without AP 10–701, the most common unsolicited AEs were infections (24/48, 50%) and gastrointestinal disorders (10/48, 21%). Upper respiratory tract infections accounted for most (17/24, 71%) of the infections, whereas abdominal pain, toothache, and dyspepsia were the most common gastrointestinal AEs. In this group, 19 of 48 (40%) participants experienced moderate unsolicited AEs, compared to 2 of 12 (17%) in the comparator group.

## Clinical laboratory AEs

Overall, 10 participants experienced 13 mild or moderate clinical laboratory abnormalities following vaccination (**S3 Table**): 8 participants vaccinated with *Sm*-TSP-2/Alhydrogel with or without AP 10–701 experienced 9 events (2 mild increases in WBC count, 1 moderate increase in WBC count, 2 mild increases in platelets, 2 mild increases in ALT, 1 moderate increase in ALT, 1 mild increase in creatinine) whereas 2 comparator recipients experienced 4 events (3 mild increases in WBC, 1 mild increase in platelets). All abnormal laboratory results were assessed as being unrelated to study vaccine except for one mild, asymptomatic vaccine-related increase in WBC count at study day 120 (7 days after the third vaccination) in a participant vaccinated with 30 μg *Sm*-TSP-2/Alhydrogel with AP 10–701; the duration of this increase was not documented since a follow-up CBC was not performed due to the lack of symptoms and minimal increase in WBC.

## Immunogenicity

**ELISA quality control statistics.**   The $R^2$s for the 4-PL SCCs of SRS generated from 16 ELISA plates ranged from 0.9942 to 0.9985 (**S1A Fig**), indicating a good dose-response between the dilutions and the ODs for the 4-PL log regression model. After the SCCs were linearized using the fully specified logit-log model, the non-significant ANOVA test result ($p>0.9999$, **S1B Fig**) indicated the SCCs were parallel.

**Total IgG responses.**   Several participants across all vaccine groups had detectable, albeit low, levels of total IgG and IgG subclass antibodies to *Sm*-TSP-2 at baseline; however, there were no significant differences in these levels between groups (**Table 2**). Geometric mean antigen-specific IgG levels did not vary significantly from baseline in the comparator vaccine group over the course of the study. In contrast, in those vaccinated with *Sm*-TSP-2/Alhydrogel, peak post-vaccination antibody levels were progressively higher with successive vaccinations in all but the 10 μg *Sm*-TSP-2/Alhydrogel group.

In the per-protocol immunogenicity group, maximum anti-*Sm*-TSP-2 IgG antibody levels in participants vaccinated with *Sm*-TSP-2/Alhydrogel were observed at study day 127, 2 weeks following the third dose, with 2 exceptions: in the 10 μg *Sm*-TSP-2/Alhydrogel group, the geometric mean level (GML) was similar at all time points, indicating a lack of response, while in the 100 μg *Sm*-TSP-2/Alhydrogel group, the peak in IgG was observed at study day 203 (**Fig 4**; individual graphs of longitudinal IgG responses for each vaccine group are shown in **S2 Fig**).

**Table 2. Anti-*Sm*-TSP-2 total IgG geometric mean levels (Arbitrary Units) and seroresponse results, by time point and vaccine group (per-protocol immunogenicity population).**

| Time Point | | 10μg *Sm*-TSP-2/ Alhydrogel (N = 8) | 10μg *Sm*-TSP-2/ Alhydrogel with AP 10–701 (N = 8) | 30μg *Sm*-TSP-2/ Alhydrogel (N = 8) | 30μg *Sm*-TSP-2/ Alhydrogel with AP 10–701 (N = 8) | 100μg *Sm*-TSP-2/ Alhydrogel (N = 8) | 100μg *Sm*-TSP-2/ Alhydrogel with AP 10–701 (N = 8) | Euvax B Vaccine (N = 12) |
|---|---|---|---|---|---|---|---|---|
| **Day 1 (Vax #1)** | n | 8 | 8 | 8 | 8 | 8 | 8 | 12 |
| GML (95% CI) | | 2.2 (1.0, 4.6) | 2.3 (1.0, 5.2) | 1.7 (1.0, 3.1) | 2.0 (0.9, 4.3) | 2.6 (1.3, 5.2) | 2.4 (1.2, 4.9) | 3.7 (2.2, 6.2) |
| GMFR (95% CI) | | - | - | - | - | - | - | - |
| Number (%) Responders[a] | | - | - | - | - | - | - | - |
| **Day 15** | n | 8 | 8 | 8 | 8 | 8 | 8 | 12 |
| GML (95% CI) | | 4.3 (0.7, 28.3) | 2.6 (0.9, 7.3) | 2.6 (1.0, 7.0) | 3.3 (1.0, 11.0) | 4.1 (1.3, 12.7) | 7.3 (3.5, 15.2) | 3.2 (1.9, 5.5) |
| GMFR (95% CI) | | 2.0 (0.5, 8.1) | 1.2 (0.9, 1.4) | 1.5 (0.7, 3.0) | 1.7 (0.8, 3.4) | 1.6 (0.9, 3.0) | 3.0 (1.3, 6.9) | 0.9 (0.7, 1.1) |
| Number (%) Responders[a] | | 1 (13) | 0 | 1 (13) | 1 (13) | 1 (13) | 3 (38) | 0 |
| **Day 57 (Vax #2)** | n | 8 | 8 | 8 | 8 | 8 | 8 | 12 |
| GML (95% CI) | | 4.1 (0.6, 26.2) | 3.9 (1.2, 12.4) | 3.1 (1.2, 7.9) | 2.8 (1.0, 7.7) | 3.0 (1.2, 7.7) | 5.9 (3.1, 11.1) | 36 (2.2, 5.9) |
| GMFR (95% CI) | | 1.8 (0.4, 7.7) | 1.7 (0.7, 4.1) | 1.8 (0.9, 3.7) | 1.4 (0.9, 2.2) | 1.2 (0.6, 2.1) | 2.4 (1.1, 5.3) | 1.0 (0.9, 1.1) |
| Number (%) Responders | | 1 (13) | 1 (13) | 2 (25) | 0 | 0 | 3 (38) | 0 |
| **Day 71** | n | 8 | 7 | 8 | 8 | 8 | 7 | 12 |
| GML (95% CI) | | 3.4 (0.8, 14.2) | 2.9 (1.2, 7.3) | 2.6 (1.0, 7.0) | 7.3 (2.6, 20.4) | 6.3 (2.2, 18.3) | 11.6 (3.8, 35.6) | 3.6 (2.3, 5.8) |
| GMFR (95% CI) | | 1.5 (0.6, 4.0) | 1.3 (0.7, 2.7) | 1.5 (0.7, 3.1) | 3.7 (1.5, 9.1) | 2.4 (1.2, 4.9) | 4.3 (1.9, 9.9) | 1.0 (0.9, 1.1) |
| Number (%) Responders | | 1 (13) | 1 (14) | 1 (13) | 4 (50) | 2 (25) | 4 (57) | 0 |
| **Day 113 (Vax #3)** | n | 8 | 6 | 8 | 8 | 8 | 7 | 12 |
| GML (95% CI) | | 3.0 (0.9, 10.6) | 3.7 (0.9, 15.0) | 2.6 (1.0, 6.9) | 3.5 (1.2, 9.9) | 5.5 (2.2, 13.4) | 10.8 (3.8, 30.3) | 3.4 (2.2, 5.2) |
| GMFR (95% CI) | | 1.4 (0.6, 3.0) | 1.5 (0.8, 2.7) | 1.5 (0.8, 3.1) | 1.8 (1.0, 3.2) | 2.1 (1.3, 3.4) | 4.1 (1.6, 10.0) | 0.9 (0.8, 1.1) |
| Number (%) Responders | | 1 (13) | 1 (17) | 1 (13) | 1 (13) | 1 (13) | 2 (29) | 0 |
| **Day 127** | n | 8 | 6 | 7 | 8 | 8 | 7 | 12 |
| GML (95% CI) | | 3.4 (1.0, 11.3) | 8.3 (3.1, 22.2) | 7.3 (2.3, 23.1) | 10.0 (5.1, 19.8) | 6.1 (1.6, 23.6) | 25.0 (5.1, 123.1) | 3.8 (2.2, 6.7) |
| GMFR (95% CI) | | 1.5 (0.7, 3.4) | 3.4 (2.1, 5.5) | 3.4 (1.1, 10.3) | 5.1 (3.0, 8.7) | 2.4 (0.8, 6.9) | 9.4 (3.4, 25.7) | 1.0 (0.8, 1.4) |
| Number (%) Responders | | 1 (13) | 2 (33) | 2 (25) | 4 (50) | 2 (25) | 6 (86) | 1 (8) |
| **Day 203** | n | 8 | 6 | 7 | 7 | 7 | 7 | 12 |
| GML (95% CI) | | 3.2 (1.2, 8.9) | 3.8 (1.0, 14.8) | 3.8 (1.3, 10.8) | 5.9 (2.3, 15.2) | 7.2 (3.3, 15.4) | 22.6 (11.9, 42.9) | 3.5 (2.2, 5.4) |
| GMFR (95% CI) | | 1.5 (0.8, 2.9) | 1.5 (0.8, 3.1) | 1.9 (0.9, 4.2) | 2.8 (1.5, 5.3) | 3.1 (1.3, 7.2) | 8.5 (4.3, 16.8) | 0.9 (0.8, 1.2) |
| Number (%) Responders | | 2 (25) | 1 (17) | 2 (25) | 2 (29) | 3 (43) | 5 (71) | 0 |
| **Day 293** | n | 8 | 6 | 7 | 7 | 6 | 7 | 12 |

*(Continued)*

**Table 2.** (Continued)

| Time Point | | 10μg *Sm*-TSP-2/ Alhydrogel (N = 8) | 10μg *Sm*-TSP-2/ Alhydrogel with AP 10–701 (N = 8) | 30μg *Sm*-TSP-2/ Alhydrogel (N = 8) | 30μg *Sm*-TSP-2/ Alhydrogel with AP 10–701 (N = 8) | 100μg *Sm*-TSP-2/ Alhydrogel (N = 8) | 100μg *Sm*-TSP-2/ Alhydrogel with AP 10–701 (N = 8) | Euvax B Vaccine (N = 12) |
|---|---|---|---|---|---|---|---|---|
| GML (95% CI) | | 3.3 (1.2, 8.8) | 3.5 (0.9, 13.0) | 3.2 (1.3, 8.3) | 4.2 (1.5, 11.8) | 4.8 (2.3, 10.1) | 16.2 (8.4, 31.0) | 3.7 (2.4, 5.8) |
| GMFR (95% CI) | | 1.5 (0.8, 2.9) | 1.4 (0.8, 2.6) | 1.7 (0.9, 3.2) | 2.0 (1.1, 3.8) | 2.6 (1.1, 5.7) | 6.1 (2.9, 12.6) | 1.0 (0.8, 1.3) |
| Number (%) Responders | | 2 (25) | 1 (17) | 1 (13) | 1 (14) | 3 (50) | 4 (57) | 0 |
| **Day 478** | n | 6 | 6 | 7 | 6 | 7 | 7 | 10 |
| GML (95% CI) | | 3.1 (0.9, 10.7) | 3.4 (1.0, 12.1) | 2.7 (1.0, 7.1) | 4.1 (1.3, 12.4) | 4.2 (2.3, 7.8) | 11.4 (6.2, 20.9) | 3.8 (2.4, 6.0) |
| GMFR (95% CI) | | 1.4 (0.7, 2.9) | 1.4 (0.8, 2.5) | 1.4 (0.7, 2.7) | 1.8 (0.8, 3.7) | 1.8 (0.8, 3.9) | 4.1 (2.2, 7.7) | 0.9 (0.7, 1.3) |
| Number (%) Responders | | 0 | 1 (17) | 1 (13) | 1 (17) | 2 (29) | 4 (57) | 0 |

Note: N = Number of participants in the per-protocol immunogenicity population. n = Number of participants with results at the given timepoint. CI = confidence interval. GML = Geometric mean level. SD = standard deviation. GMFR = Geometric mean fold rise in antibody level compared to baseline.

[a]Participants were considered to have a positive antibody response if the fold rise from baseline was at least 4.0.

While significant differences in geometric mean IgG levels were not observed between groups vaccinated with *Sm*-TSP-2/Alhydrogel with or without AP 10–701, groups given *Sm*-TSP-2/ Alhydrogel with AP 10–701 tended to have higher GMLs of antigen-specific IgG antibody. IgG antibodies fell to low levels by study day 478 in all groups except the 100 μg *Sm*-TSP-2/ Alhydrogel with AP 10–701 group, which had a GML of 11.4 AU (95% CI, 6.2–20.9) at this time point, 12 months after the third vaccination, compared to 2.4 AU (95% CI, 1.2–4.9) at baseline.

Only 1 comparator vaccine recipient developed a significant rise in anti-*Sm*-TSP-2 IgG antibodies during the study, and at only 1 time point: this participant had a ≥4-fold rise in IgG (from 4.8 to 21.5 AU) from the baseline to the study day 127 visits. Of note, this participant tested negative for *S. mansoni* by Kato Katz at both baseline and study day 293, so this rise in IgG cannot be ascribed to infection. In contrast, the proportions of participants with ≥4-fold increases in IgG antibodies after vaccination (*i.e.*, seroresponders) were high in the 30 μg *Sm*-TSP-2/Alhydrogel with AP 10–701 group and were highest in the 100 μg *Sm*-TSP-2/Alhydrogel with AP 10–701 group (**Table 2**), with these responses peaking on study day 127, 2 weeks after the third dose of vaccine. At this time point, 4/8 (50%) of 30 μg *Sm*-TSP-2/Alhydrogel with AP 10–701 recipients and 6/7 (86%) of 100 μg *Sm*-TSP-2/Alhydrogel with AP 10–701 recipients were seroresponders, respectively. Proportions of seroresponders fell in all groups for the remainder of the study until day 478 except in the 100 μg plus AP 10–701 group, in which 57% of participants (4 of 7) continued to have IgG levels that were ≥4-fold higher than baseline.

There were significantly more (86%) antigen-specific IgG responders in the 100 μg with AP 10–701 group than in the comparator vaccine (8%) group (*p* = 0.002) at study day 127 (**Fig 5**). However, proportions of responders in the other vaccine groups were not significantly different from the comparator group. Nevertheless, the results suggest a dose-response relationship for *Sm*-TSP-2/Alhydrogel when administered with AP 10–701. At study day 127, seroresponse frequencies were 13%, 29%, and 25%, respectively, for the 10 μg, 30 μg, and 100 μg *Sm*-TSP-2/

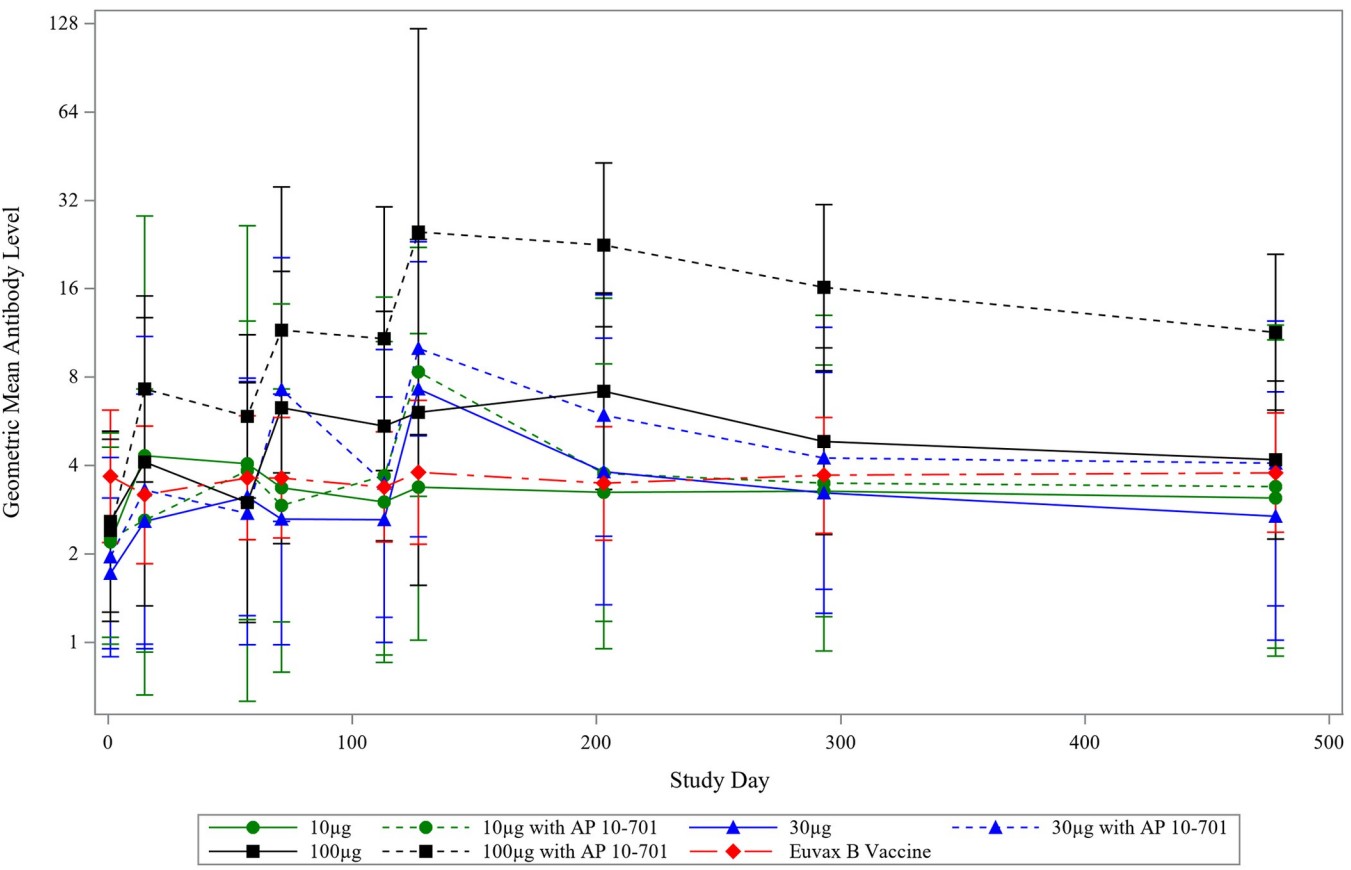

Euvax B Vaccine subjects are pooled across cohorts.

**Fig 4. Geometric mean anti-*Sm*-TSP-2 IgG levels over time by vaccine group, as measured by ELISA (Arbitrary Units).** Per-protocol immunogenicity population. Vaccinations were administered on study days 1, 57, and 113. Error bars represent 95% confidence intervals. Euvax B hepatitis B recipients were pooled across cohorts.

Alhydrogel without AP 10–701 groups. For the groups given 10 μg, 30 μg, and 100 μg *Sm*-TSP-2/Alhydrogel with AP 10–701, the frequencies of responders were 33%, 50%, and 86%, respectively. Cochran Armitage tests for linear trend with increasing dose of *Sm*-TSP-2 yielded *p*-values of 0.339 for the non-AP 10–701 groups and 0.001 for the *Sm*-TSP-2/Alhydrogel plus AP 10–701 groups, including the comparator vaccine group in each test at a dose level of zero.

When assessed by ANCOVA, there were no significant differences in adjusted mean log IgG antibody responses at study day 127 between those who were vaccinated with *Sm*-TSP-2/Alhydrogel (with or without AP 10–701) and who were negative for *S. mansoni* on fecal exam during screening (n = 40) and those who were positive (n = 4) in the per-protocol immunogenicity population (*p* = 0.062). However, the number of participants who were positive for *S. mansoni* by Kato Katz fecal smear was low. Graphs of individual longitudinal IgG responses are shown by vaccine group and *S. mansoni* infection status (both at baseline and at study day 293) in **S2 Fig**.

## IgG Subclass responses

Antigen-specific IgG1 antibodies were induced by vaccination with *Sm*-TSP-2/Alhydrogel with or without AP 10–701 in a pattern similar to total anti-*Sm*-TSP-2 IgG responses (**Fig 6A**). Graphs of individual longitudinal IgG1 subclass responses are shown for each vaccine group

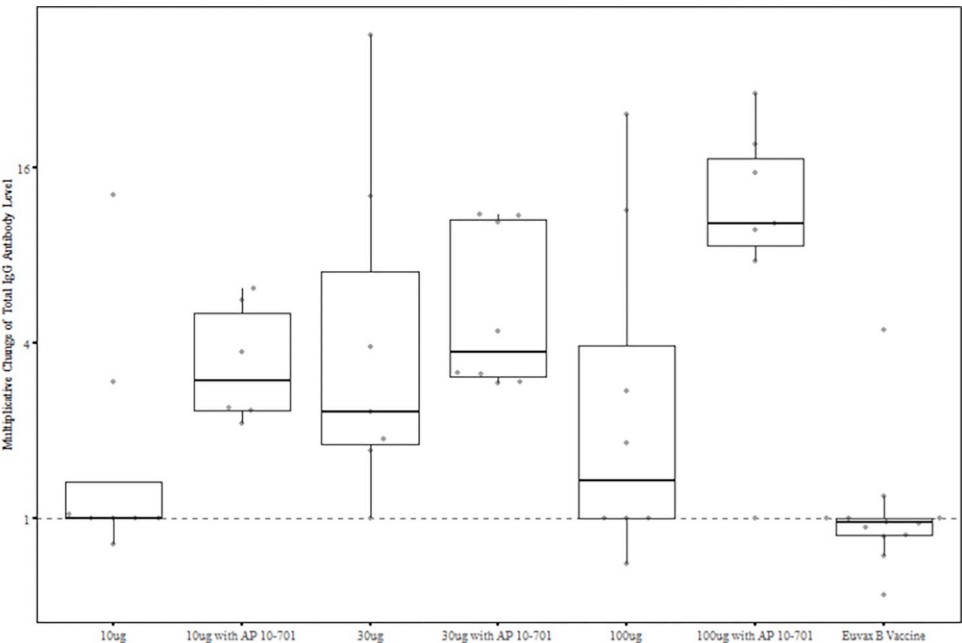

**Fig 5. Fold change from baseline in anti-*Sm*-TSP-2 IgG levels at study day 127 by vaccine group.** Per-protocol immunogenicity population. Vaccinations were administered on study days 1, 57, and 113. Error bars represent 95% confidence intervals. Euvax B hepatitis B recipients were pooled across cohorts.

in **S3 Fig**. Increases in IgG1 levels were highest on study day 127, 2 weeks after the third vaccination, for most vaccine groups, with the highest levels seen in the 100 µg *Sm*-TSP-2/Alhydrogel with AP 10–701 group. Although not statistically significant due to small group sizes, IgG1 responses were higher with increasing doses of *Sm*-TSP-2 and in the AP 10–701 groups compared to the non-AP 10–701 groups. In the 100 µg *Sm*-TSP-2/Alhydrogel with and without AP 10–701 groups, 5/7 (71%) and 6/8 (75%) of study participants, respectively, were

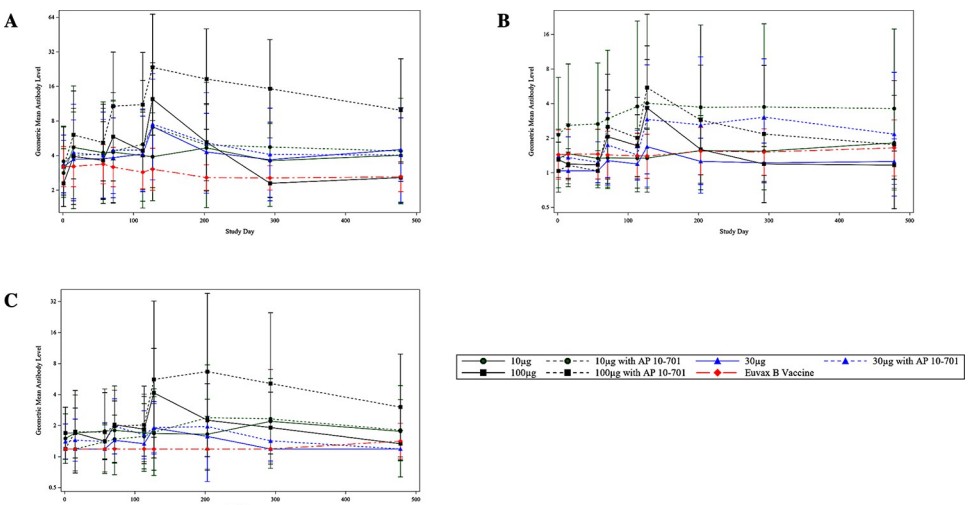

**Fig 6. Geometric mean anti-*Sm*-TSP-2 IgG subclass responses over time by vaccine group, as measured by ELISA (Arbitrary Units): A) IgG1, B) IgG3, C) IgG4.** (Per-protocol immunogenicity population). Note: Vaccinations were administered on study days 1, 57, and 113. Error bars represent 95% confidence intervals. Euvax B hepatitis B recipients were pooled across cohorts.

seroresponders on study day 127 compared to none in the comparator vaccine group
($p$ = 0.002 and 0.001, respectively).

The patterns of IgG3 and IgG4 subclass responses were similar (**Fig 6B and 6C**). Compared to IgG1 responses, increases in levels were smaller following vaccination, but peaked on study day 127. For IgG3, 5/7 (71%) in the 100 μg *Sm*-TSP-2/Alhydrogel with AP 10–701 group and 3/8 (38%) in the 100 μg *Sm*-TSP-2/Alhydrogel without AP 10–701 group were seroresponders for this antibody subclass compared to 0/12 in the comparator group (p = 0.002 and 0.049, respectively); whereas for IgG4, only the 100 μg *Sm*-TSP-2/Alhydrogel with AP 10–701 group had a significantly greater rate of seroresponse compared to the comparator group (43% *vs*. 0%, p = 0.036). Graphs of individual longitudinal IgG3 and IgG4 subclass responses are shown by vaccine group in **S4** and **S5** **Figs**, respectively.

## Discussion

The results of this Phase 1b, observer-blind, randomized, dose-escalation clinical trial of a candidate recombinant protein vaccine against *S. mansoni* are summarized herein. When administered to healthy young adults who reside in an area of Brazil with active *S. mansoni* transmission, the *Sm*-TSP-2/Alhydrogel schistosomiasis vaccine was found to be safe and well tolerated. Mild injection site tenderness and pain, as well as mild headache, were the most common solicited reactions. Solicited injection site and systemic reactions were short-lived, lasting only a median of 2.5 and <1 days, respectively. Injection site tenderness and pain were reported significantly more frequently among participants receiving *Sm*-TSP-2/Alhydrogel with or without AP 10–701 than participants vaccinated with the comparator hepatitis B vaccine, whereas rates of systemic symptoms were similar between groups. The frequencies of clinical laboratory events judged to be associated with vaccination were low, and these events were mild, transient, and clinically insignificant.

The safety results of the current study conducted in healthy adults resident in a region endemic for *S. mansoni* are similar to those of the first-in-humans Phase 1a trial of this vaccine conducted in schistosomiasis-naïve adults in Houston, Texas [17]. However, of note, the finding of a possible trend for the occurrence of mild fever among recipients of *Sm*-TSP-2/Alhydrogel with AP 10–701 (referred to as GLA-AF in the previous study) was not observed in the study conducted in Brazil.

Baseline pre-vaccinations levels of IgG to *Sm*-TSP-2 were detected in several participants across vaccine groups, which is likely secondary to previous infection or exposure to the parasite [28]. These baseline IgG levels to *Sm*-TSP-2 are consistent with previous studies in the same region of Brazil, where chronically infected individuals had low levels of IgG to this antigen compared to putatively resistant individuals, an observation suggesting that natural infection does not induce the robust antibody responses to this transmembrane protein that we hypothesize are required for protection. Encouragingly, the presence of pre-existing antibody responses to the *Sm*-TSP-2 antigen did not appear to interfere with the humoral response to vaccination, since significant fold-increases in IgG were observed particularly in those vaccinated with 30 or 100 μg of antigen, to a degree equivalent to or exceeding that previously reported in schistosomiasis-naïve individuals [17]. Since *S. mansoni* is the only schistosome species that infects humans in South America, background immune responses to other schistosome species would not be expected to impact the assessment or interpretation of vaccine immunogenicity as might be the case in other regions such as sub-Saharan Africa, where infection with species other than *S. mansoni* might modify the immune response to vaccination.

Formulation- and dose-dependent increases in IgG antibodies against the *Sm*-TSP-2 vaccine antigen with adjuvant were observed. The highest GMLs of anti-*Sm*-TSP-2 IgG were seen

in participants who received 30 µg or 100 µg of *Sm*-TSP-2/Alhydrogel when co-administered with the AP 10–701 adjuvant. Although significant differences in the GML of IgG between groups vaccinated with *Sm*-TSP-2/Alhydrogel with or without AP 10–701 were not observed, groups given *Sm*-TSP-2/Alhydrogel with AP 10–701 had higher GMLs of antigen-specific IgG antibody compared to those who received *Sm*-TSP-2/Alhydrogel without AP 10–701, suggesting a positive adjuvant effect of this TLR-4 agonist.

Across dose concentrations and formulations, the highest antigen-specific IgG responses were generally noted at study day 127, 2 weeks following receipt of the third dose of *Sm*-TSP-2/Alhydrogel. Similarly, there were significantly more antigen-specific IgG seroresponders in the 100 µg with AP 10–701 group than in the comparator group at this time point. Although the proportions of seroresponders in the other vaccine groups were not significantly different from the comparator group, the results are suggestive of a dose-response relationship for *Sm*-TSP-2/Alhydrogel when administered with AP 10–701, in that the proportion of responders was higher when *Sm*-TSP-2/Alhydrogel was administered with AP 10–701 at each of the 3 *Sm*-TSP-2 dose levels tested.

Although levels of antibody declined in the lower dose groups in the 12 months following vaccinations, it is encouraging to note that in the 100 µg *Sm*-TSP-2/Alhydrogel with AP 10–701 group, IgG levels remained significantly above baseline until the end of the study. Given the putative antibody-mediated mechanism of this vaccine, sustained levels of antibody are presumed to be important for long-term efficacy. Future studies may be needed to explore if additional booster doses will elicit higher IgG levels that may be required to maintain protection.

Based partly on the immunogenicity and safety results of this trial, 100 µg *Sm*-TSP-2/Alhydrogel with AP 10–701 is the dose and formulation currently being tested in a Phase 2 proof-of-efficacy trial in Ugandan adults. In this study, volunteers with detectable eggs of *S. mansoni* on microscopic examination of a fecal sample are treated with praziquantel before being vaccinated with the *Sm*-TSP-2 vaccine or an active comparator. The primary endpoint will be the rate of re-infection over an 18-month period of follow-up.

Vaccines targeting other helminths such as hookworm have triggered IgE-mediated hypersensitivity reactions when tested in individuals who had previously been exposed to the pathogen [18]. Encouraging for the continued development of *Sm*-TSP-2/Alhydrogel, only 1 of 126 volunteers who were screened for IgE antibodies to *Sm*-TSP-2 during eligibility assessment for the current study tested positive using a qualified indirect ELISA, and even this individual had an antigen-specific IgE level only slightly above the limit of detection. Furthermore, no vaccine-related hypersensitivity reactions were observed after any of the vaccinations in the clinical trial reported here.

In the first clinical trial of *Sm*-TSP-2/Alhydrogel conducted in a population with endemic *S. mansoni* infection, several lines of evidence converge to justify advancing this recombinant version of the extracellular loop of the *Sm*-TSP-2 tegumentary transmembrane protein as a vaccine candidate. First, *Sm*-TSP-2 was found to be as safe and well tolerated as it was when administered in a first-in-humans Phase 1a trial to naïve individuals in the USA, a finding that is especially important given previous experience in the same area of Brazil where vaccinations with the hookworm excretory/secretory protein *Na*-ASP-2, which is similarly accessible (and not a 'hidden' antigen) to the host as *Sm*-TSP-2, resulted in immediate-type hypersensitivity reactions. Second, in both the USA and Brazil Phase 1 studies, *Sm*-TSP-2/Alhydrogel elicited robust IgG responses in a dose-dependent manner and consisting primarily of IgG1, which mirrors the predominant humoral immune response to *Sm*-TSP-2 observed in putatively resistant individuals resident in the same *S. mansoni* endemic area of Brazil as the trial reported herein [16]. Third, although IgG to *Sm*-TSP-2 was observed at baseline in several individuals

in the current study, an observation reported in only 1 individual in the Phase 1a study in the United States, significant increases in IgG were seen in this population, indicating that low levels of pre-existing antibodies do not interfere with vaccine-induced responses.

Although *Sm*-TSP-2 targets just one of the two most prevalent schistosome species, it is possible that this antigen may also induce cross-reactive antibodies or other immune responses to other schistosome species, particularly *S. haematobium*. Although no empiric cross-reactivity studies have been conducted to date, the amino acid sequence similarities between the *S. haematobium* tetraspanins, including *Sh*-TSP-2, and their *S. mansoni* homologs ranges from 71–93% when entire open reading frames are compared and 70–84% when only the large extracellular loop regions are considered [29]. Therefore, it is plausible that IgG antibodies induced by the *Sm*-TSP-2/Alhydrogel vaccine will cross-react with *Sh*-TSP-2 and potentially protect against this schistosome species.

*Sm*-TSP-2 is one of several ubiquitous integral tetraspanin membrane proteins, and antibodies targeting it are hypothesized to disrupt the epithelial syncytium that forms the schistosome tegument, a dynamic outer layer maintained and organized by abundant proteins such as *Sm*-TSP-2. The abundant levels of the IgG subclasses induced by the recombinant protein in this Phase 1b trial are encouraging for this vaccine due to their critical role in antibody-dependent cell-mediated cytotoxicity, which is the proposed mechanism for disrupting the schistosome tegument [30]. Although the functionality of the anti-*Sm*-TSP-2 antibodies induced in participants in this study was not assessed, the immunogenicity and safety results of the Phase 1b trial reported in this manuscript are nevertheless encouraging. Larger studies need to be conducted in populations with high levels of exposure to *S. mansoni* and other *Schistosoma* species to assess the efficacy of the vaccine in preventing infection and to evaluate the safety and immunogenicity in children, who will be the principal target of a licensed vaccine for schistosomiasis [12].

## Supporting information

**S1 Consort checklist. Consort checklist**
(PDF)

**S1 Fig. ELISA quality control**: (A) 16 standard calibration curves (SCCs) generated from each ELISA plate were plotted along a four-parameter logistic log scale, where the X-axis represents the log of the dilution of SRS in Arbitrary Units (AU) and the Y-axis its Optical Density (OD) at 492nm. SRS = Standard Reference Serum; (B) Parallelism test: linearization of the 16 SCCs shown in panel (A) using a logit-log scale. The X-axis represents the log of the dilution and the Y-axis represents the fully specified logit of $OD_{492nm}$. Tests of parallelism were performed using an ANOVA test, which indicated no significant departure from parallelism ($p>0.9999$).
(TIF)

**S2 Fig. Individual study participant anti-*Sm*-TSP-2 ELISA IgG values over time, by *S. mansoni* infection status at baseline and at study day 293:** (A) 10 µg *Sm*-TSP-2/Alhydrogel; (B) 10 µg *Sm*-TSP-2/Alhydrogel with AP 10–701; (C) 30 µg *Sm*-TSP-2/Alhydrogel; (D) 30 µg *Sm*-TSP-2/Alhydrogel with AP 10–701; (E) 100 µg *Sm*-TSP-2/Alhydrogel; (F) 100 µg *Sm*-TSP-2/Alhydrogel with AP 10–701; (G) Euvax Hepatitis B vaccine.
(TIF)

**S3 Fig. Individual study participant anti-*Sm*-TSP-2 ELISA IgG1 values over time:** (A) 10 µg *Sm*-TSP-2/Alhydrogel; (B) 10 µg *Sm*-TSP-2/Alhydrogel with AP 10–701; (C) 30 µg *Sm*-TSP-2/Alhydrogel; (D) 30 µg *Sm*-TSP-2/Alhydrogel with AP 10–701; (E) 100 µg *Sm*-TSP-2/

Alhydrogel; (F) 100 µg *Sm*-TSP-2/Alhydrogel with AP 10–701; (G) Euvax Hepatitis B vaccine. (TIF)

**S4 Fig. Individual study participant anti-*Sm*-TSP-2 ELISA IgG3 values over time:** (A) 10 µg *Sm*-TSP-2/Alhydrogel; (B) 10 µg *Sm*-TSP-2/Alhydrogel with AP 10–701; (C) 30 µg *Sm*-TSP-2/Alhydrogel; (D) 30 µg *Sm*-TSP-2/Alhydrogel with AP 10–701; (E) 100 µg *Sm*-TSP-2/ Alhydrogel; (F) 100 µg *Sm*-TSP-2/Alhydrogel with AP 10–701; (G) Euvax Hepatitis B vaccine. (TIF)

**S5 Fig. Individual study participant anti-*Sm*-TSP-2 ELISA IgG4 values over time:** (A) 10 µg *Sm*-TSP-2/Alhydrogel; (B) 10 µg *Sm*-TSP-2/Alhydrogel with AP 10–701; (C) 30 µg *Sm*-TSP-2/Alhydrogel; (D) 30 µg *Sm*-TSP-2/Alhydrogel with AP 10–701; (E) 100 µg *Sm*-TSP-2/ Alhydrogel; (F) 100 µg *Sm*-TSP-2/Alhydrogel with AP 10–701; (G) Euvax Hepatitis B vaccine. (TIF)

**S1 Table. *S. mansoni* egg counts per gram of feces for participants who tested positive at baseline or study day 293, as measured by Kato Katz fecal thick smear.** (DOCX)

**S2 Table. Summary statistics for the duration (days) of solicited adverse events, by vaccine group.** (DOCX)

**S3 Table. Listing of clinical laboratory adverse events experienced by study participants.** (DOCX)

# Acknowledgments

We thank the study participants for their cooperation throughout the trial; the study team in Brazil, especially Cássia Senra, Simone Pinto, Stella Sobrinho, Ana Raquel Godoy, Fernanda Gambogi, and Roberta Rodrigues; Christiane Correa Rodrigues Cimini for being the independent safety monitor; and, Arthur Clinton White, Jr., Thomas Richie, and Kirsten Lyke for serving as members of the independent study Safety Monitoring Committee.

# Author Contributions

**Conceptualization:** David J. Diemert, Rodrigo Correa-Oliveira, Shital M. Patel, Gregory A. Deye, Maria Elena Bottazzi, Peter J. Hotez, Wendy A. Keitel, Jeffrey Bethony, Robert L. Atmar.

**Data curation:** David J. Diemert, Shirley Galbiati, Jessie K. Kennedy, Guangzhao Li, Lara Hoeweler.

**Formal analysis:** David J. Diemert, Shirley Galbiati, Jessie K. Kennedy, Jordan S. Lundeen, Guangzhao Li, Wendy A. Keitel, Jeffrey Bethony, Robert L. Atmar.

**Funding acquisition:** David J. Diemert, Shital M. Patel, Maria Elena Bottazzi, Peter J. Hotez, Hana M. El Sahly, Wendy A. Keitel, Jeffrey Bethony, Robert L. Atmar.

**Investigation:** David J. Diemert, Carlo Geraldo Fraga, Frederico Talles, Marcella Rezende Silva, Maria Flavia Gazzinelli, Lara Hoeweler.

**Methodology:** David J. Diemert, Rodrigo Correa-Oliveira, Shital M. Patel, Jessie K. Kennedy, Maria Flavia Gazzinelli, Maria Elena Bottazzi, Peter J. Hotez, Wendy A. Keitel, Jeffrey Bethony, Robert L. Atmar.

**Project administration:** David J. Diemert, Rodrigo Correa-Oliveira, Shital M. Patel, Gregory A. Deye, Hana M. El Sahly, Wendy A. Keitel, Jeffrey Bethony, Robert L. Atmar.

**Resources:** Gregory A. Deye.

**Supervision:** David J. Diemert, Rodrigo Correa-Oliveira, Carlo Geraldo Fraga, Frederico Talles, Marcella Rezende Silva, Shital M. Patel, Maria Flavia Gazzinelli, Jeffrey Bethony, Robert L. Atmar.

**Validation:** David J. Diemert, Guangzhao Li.

**Visualization:** David J. Diemert, Shirley Galbiati, Jessie K. Kennedy, Jordan S. Lundeen, Guangzhao Li, Jeffrey Bethony.

**Writing – original draft:** David J. Diemert, Guangzhao Li, Jeffrey Bethony, Robert L. Atmar.

**Writing – review & editing:** David J. Diemert, Rodrigo Correa-Oliveira, Carlo Geraldo Fraga, Frederico Talles, Marcella Rezende Silva, Shital M. Patel, Shirley Galbiati, Jessie K. Kennedy, Jordan S. Lundeen, Maria Flavia Gazzinelli, Guangzhao Li, Lara Hoeweler, Gregory A. Deye, Maria Elena Bottazzi, Peter J. Hotez, Hana M. El Sahly, Wendy A. Keitel, Jeffrey Bethony, Robert L. Atmar.

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
