## [Decision Letter · Decision Letter 0]

30 Dec 2022

Dear Dr. Diemert,

Thank you very much for submitting your manuscript "A Randomized, Controlled Phase 1b Trial of the Sm-TSP-2 Vaccine for Intestinal Schistosomiasis in Healthy Brazilian Adults Living in an Endemic Area" for consideration at PLOS Neglected Tropical Diseases. As with all papers reviewed by the journal, your manuscript was reviewed by members of the editorial board and by several independent reviewers. The reviewers appreciated the attention to an important topic. Based on the reviews, we are likely to accept this manuscript for publication, providing that you modify the manuscript according to the review recommendations. 

Sincerely,

Javier Sotillo

Academic Editor

Cinzia Cantacessi

Section Editor

Reviewer's Responses to Questions

**Key Review Criteria Required for Acceptance?**

**Methods**

-Are the objectives of the study clearly articulated with a clear testable hypothesis stated?

-Is the study design appropriate to address the stated objectives?

-Is the population clearly described and appropriate for the hypothesis being tested?

-Is the sample size sufficient to ensure adequate power to address the hypothesis being tested?

-Were correct statistical analysis used to support conclusions?

-Are there concerns about ethical or regulatory requirements being met?

Reviewer #1: (No Response)

Reviewer #2: Yes

**Results**

-Does the analysis presented match the analysis plan?

-Are the results clearly and completely presented?

-Are the figures (Tables, Images) of sufficient quality for clarity?

Reviewer #1: (No Response)

Reviewer #2: Mostly. See comments

**Conclusions**

-Are the conclusions supported by the data presented?

-Are the limitations of analysis clearly described?

-Do the authors discuss how these data can be helpful to advance our understanding of the topic under study?

-Is public health relevance addressed?

Reviewer #1: (No Response)

Reviewer #2: Yes

**Editorial and Data Presentation Modifications?**

Reviewer #1: (No Response)

Reviewer #2: see below

**Summary and General Comments**

Reviewer #1: This is a very well written manuscript that describes the successful completion of Phase 1b trial of the Sm-TSP-2 vaccine against schistosomiasis. First, I will like to congratulate the authors for validating the safety of the vaccines using a randomized, observer-blind, controlled phase 1b clinical trial in 60 healthy adults living in a region of Brazil with ongoing S. mansoni transmission, as well as showing that the two adjuvanted Sm-TSP-2 vaccines (+alum or +alum/AP 10-701) are minimally reactogenic, and importantly both elicited significant IgG and IgG subclass responses against the vaccine antigen after the third dose (in particular when 100 ug of the antigen was used). The outcomes of this study have led to the initiation of a Phase 2 clinical trial of this vaccine in an endemic region of Uganda.

I have few minor recommendations that might enhance the value of this study. 

Although the authors have clearly described the experimental details of the clinical trial and its outcomes (safety and immunogenicity), I recommend that the authors share some of their immunogenicity data more plainly, which may help other researchers who are considering conducting clinical trials to evaluate their own vaccine studies. Specifically, and for example, how does 4-fold increase in arbitrary units (AU) (using 1:4000 dilution of sera) translates to actual ELISA ODs; seroresponse was defined as >4-fold rise over baseline. 

If I understand it correctly, analysis of all the Standard Reference Serum (SRS) control testing (Figure S1) implies that a 10-100 AU range is associated with OD range of 0.5-3.5, while 1:4000 dilution of the SRS is about an OD of 0.5. As the antigenicity outcomes were presented only as AU (Figs 4 and 6) and fold change (Fig. 5) it is very hard to recognize the actual levels (OD at 1:4000) of the IgG and/or IgG1 and IgG3 responses that were elicited, which have made the 100ug adjuvanted vaccines more potent than the 10 or 30 ug doses (Figure 4; Table 2); what are for example the ODs of 16.2 fold AU (day 293) vs AU of 25 (day 127) of the Am-TSP-2 + alum/AP 10-701 vaccine vs. the other groups and vs. the baseline of 4-fold AU? Adding such pertinent supporting information where appropriate could enhance the value of the comparative immunogenicity studies across the 6 experimental vaccine groups. 

I also wonder whether presenting the data in the various groups separately (like in figure S2) instead a congregated format of the 7 groups (Figure 4 and Figure 6) might be more informative and show more clearly the kinetics in each group’s participants and their associated error bars. As presented, there is too much indistinguishable data in each graph.

Second, the vaccine antigen was selected “based on its unique recognition by cytophilic antibodies in putatively immune individuals living in areas of ongoing S. mansoni transmission in Brazil, and preclinical studies in which vaccination with Sm-TSP-2 protected mice following infection”. 

I wonder if a functional assay was developed that can correlate directly and specifically the increased immunogenicity with the functionality of the anti-Sm-TSP-2 IgG and cytophilic antibodies elicited by the best vaccine formulation. If there is such an assay, was it done, and if not, a discussion pointing to such an experimental gap should be considered. In reference 14, it was shown that the putatively immune individuals have significant elevated mean IgG1 and IgG3 responses (1:100, ELISA) against TSP-2 of OD ~1.5 and 1, respectively, in comparison to the chronically infected individuals. What are the optimal titers of induced TSP-2 cytophilic antibodies that can be associated indirectly with protection?

Stating simply in the discussion that the vaccine elicited IgG responses “consisting primarily of IgG1, which parallels the unique and presumably protective humoral immune response to Sm-TSP-2 observed in putatively resistant individuals resident in the same S. mansoni endemic area of Brazil” is insufficient as it is not based on direct functional evidence and therefore might be somewhat misleading that the desired protective humoral responses having a critical role in ADCC were actually elicited. 

Thirdly, it is not clear which formulation and dose(s) was selected for the Phase 2 clinical trial; Sm-TSP-2/Alhydrogelor or the Sm-TSP-2/Alhydrogel + AP 10-701, and what readout are being expected. Such statements might be instructive.

Reviewer #2: Diemert and co-authors report a well conducted Phase Ib study of Sm-TSP-2.

The following are some considerations for improving the manuscript:

Introduction:

Para 3 from line 103. The key data describing the recognition of Sm-TSP by putatively immune humans is not referenced (line 117). Reference 14 does not match the mouse challenge model.

Methods.

It appears that Ag-specific IgE was only tested for at enrolment. I would be curious if Ag-specific IgE was generated by the vaccine.

Results

Baseline parasitolgical investigation. As the authors highlight, the TSP experience with pre-sensitization has been an issue of interest. A more comprehensive reporting of the baseline parasitologic findings would be appropriate (beyond kato katz for Sm and Ag-specific IgE). A more comprehensive parasitologic evaluation is described in the methods. Was baseline schisto serology done on study subjects? Any baseline parasitologic data should be presented (even if in supplement)

Line 392 “renal dilatation” is not something I don’t recognize. Suggest omit and just state “rigid ureterorenal lithotripsy”

Clinical Lab AEs (from line 457): Actual lab values should be given in text (or in supplement)

Table 2: 

This does not display the key data well. Graphical presentation of key findings would be better. I recognize that 63 scatterplots would be excessive (!) but 7 serial scatterplots of actual Ab level would show the data better. Raw data should be presented in supplement, or a data sharing statement included.

Discussion

Do the authors predict cross-reactive responses to Sh TSP?

Do the authors have any data from animal experiments or seroepidemiologic data to infer what a likely “protective” Ab titer would be?

The authors imply that one of the formulations is going into Phase II. Which is it and what is the rationale for this decision?

PLOS authors have the option to publish the peer review history of their article (what does this mean?). If published, this will include your full peer review and any attached files.

Reviewer #1: No

Reviewer #2: No

Figure Files:

Data Requirements:

Reproducibility:

References

---

## [Decision Letter · Decision Letter 1]

12 Mar 2023

Dear Dr. Diemert,

We are pleased to inform you that your manuscript 'A Randomized, Controlled Phase 1b Trial of the Sm-TSP-2 Vaccine for Intestinal Schistosomiasis in Healthy Brazilian Adults Living in an Endemic Area' has been provisionally accepted for publication in PLOS Neglected Tropical Diseases.

Best regards,

Javier Sotillo

Academic Editor

Cinzia Cantacessi

Section Editor

Reviewer's Responses to Questions

**Key Review Criteria Required for Acceptance?**

**Methods**

-Are the objectives of the study clearly articulated with a clear testable hypothesis stated?

-Is the study design appropriate to address the stated objectives?

-Is the population clearly described and appropriate for the hypothesis being tested?

-Is the sample size sufficient to ensure adequate power to address the hypothesis being tested?

-Were correct statistical analysis used to support conclusions?

-Are there concerns about ethical or regulatory requirements being met?

Reviewer #1: (No Response)

Reviewer #2: The authors have answered my questions/concerns.

**Results**

-Does the analysis presented match the analysis plan?

-Are the results clearly and completely presented?

-Are the figures (Tables, Images) of sufficient quality for clarity?

Reviewer #1: (No Response)

Reviewer #2: The authors have answered my questions/concerns.

**Conclusions**

-Are the conclusions supported by the data presented?

-Are the limitations of analysis clearly described?

-Do the authors discuss how these data can be helpful to advance our understanding of the topic under study?

-Is public health relevance addressed?

Reviewer #1: (No Response)

Reviewer #2: The authors have answered my questions/concerns.

**Editorial and Data Presentation Modifications?**

Reviewer #1: (No Response)

Reviewer #2: nil

**Summary and General Comments**

Reviewer #1: The authors addressed the raised points satisfactorily.

Reviewer #2: The authors have answered my questions/concerns.

PLOS authors have the option to publish the peer review history of their article (what does this mean?). If published, this will include your full peer review and any attached files.

Reviewer #1: No

Reviewer #2: **Yes: **James McCarthy

---

## [Editor Report · Acceptance letter]

27 Mar 2023

Dear Dr. Diemert,

We are delighted to inform you that your manuscript, "A Randomized, Controlled Phase 1b Trial of the Sm-TSP-2 Vaccine for Intestinal Schistosomiasis in Healthy Brazilian Adults Living in an Endemic Area," has been formally accepted for publication in PLOS Neglected Tropical Diseases.

Best regards,

Shaden Kamhawi

co-Editor-in-Chief

Paul Brindley

co-Editor-in-Chief
